# Total Phenolic Content, Flavonoid Content and Antioxidant Potential of Wild Vegetables from Western Nepal

**DOI:** 10.3390/plants8040096

**Published:** 2019-04-11

**Authors:** Sushant Aryal, Manoj Kumar Baniya, Krisha Danekhu, Puspa Kunwar, Roshani Gurung, Niranjan Koirala

**Affiliations:** 1Department of Pharmacy, Universal College of Medical Sciences, Tribhuvan University, Bhairahawa, Rupandehi 32900, Nepal; baniyamanoj76@gmail.com (M.K.B.); kdanekhu@gmail.com (K.D.); puspakunwar92@gmail.com (P.K.); 2Department of Pharmacy, Shree Medical and Technical College, Purbanchal University, Bharatpur, Chitwan 44200, Nepal; grg.rosni1990@gmail.com; 3Department of Natural Products Research, Dr. Koirala Research Institute for Biotechnology and Biodiversity, Kathmandu 44600, Nepal

**Keywords:** phenolic content, flavonoid content, antioxidant activity, wild leafy plants

## Abstract

Eight selected wild vegetables from Nepal (*Alternanthera sessilis*, *Basella alba*, *Cassia tora*, *Digera muricata*, *Ipomoea aquatica*, *Leucas cephalotes*, *Portulaca oleracea* and *Solanum nigrum*) were investigated for their antioxidative potential using 2,2-dyphenyl-1-picrylhydrazyl (DPPH) scavenging, hydrogen peroxide (H_2_O_2_), ferric reducing antioxidant power (FRAP), and ferric thiocyanate (FTC) methods. Among the selected plant extracts *C. tora* displayed the highest DPPH radical scavenging activity with an IC_50_ value 9.898 μg/mL, whereas *A. sessilis* had the maximum H_2_O_2_ scavenging activity with an IC_50_ value 16.25 μg/mL—very close to that of ascorbic acid (16.26 μg/mL). *C. tora* showed the highest absorbance in the FRAP assay and the lowest lipid peroxidation in the FTC assay. A methanol extract of *A. sessilis* resulted in the greatest phenolic content (292.65 ± 0.42 mg gallic acid equivalent (GAE)/g) measured by the Folin–Ciocalteu reagent method, while the smallest content was recorded for *B. alba* (72.66 ± 0.46 GAE/g). The greatest flavonoid content was observed with extracts of *P. oleracea* (39.38 ± 0.57 mg quercetin equivalents (QE)/g) as measured by an aluminium chloride colorimetric method, while the least was recorded for *I. aquatica* (6.61 ± 0.42 QE/g). There was a strong correlation between antioxidant activity with total phenolic (DPPH, R^2^ = 0.75; H_2_O_2_, R^2^ = 0.71) and total flavonoid content (DPPH, R^2^ = 0.84; H_2_O_2_, R^2^ = 0.66). This study demonstrates that these wild edible leafy plants could be a potential source of natural antioxidants.

## 1. Introduction

The generation of highly reactive oxygen species (ROS) with a lone unpaired electron induce oxidative stress and plays a key role in the pathogenesis of numerous physiological conditions, including cellular injury, aging, cancer, and hepatic, neurodegenerative, cardiovascular and renal disorders [1,2]. Environmental pollutants, radiation, chemicals, toxins, deep-fried foods and spicy foods, as well as physical stress are responsible for generating reactive oxygen radicals that induce the formation of abnormal proteins, leading to the depletion of antioxidants in the immune system [3]. There are a number of endogenous antioxidant enzymes, such as glutathione peroxidase, catalase and superoxide dismutase, which are capable of deactivating free radicals and therefore maintaining optimal cellular functions [4]. However, endogenous antioxidants may not be sufficient to maintain optimal cellular functions under increased oxidative stress and therefore dietary antioxidants may be necessary [5].

In recent decades, phenolic- and flavonoid-rich natural diets with antioxidant activity have fostered interest in nutrition and food science [6]. Natural phenolic and flavonoid compounds are plant secondary metabolites that hold an aromatic ring bearing at least one hydroxyl group [7]. Phenolic compounds are good electron donors because their hydroxyl groups can directly contribute to antioxidant action [8]. Furthermore, some of them stimulate the synthesis of endogenous antioxidant molecules in the cell [9]. According to multiple reports in the literature, phenolic compounds exhibit free radical inhibition, peroxide decomposition, metal inactivation or oxygen scavenging in biological systems and prevent oxidative disease burden [10].

Natural antioxidants from leafy vegetables play a vital role in protecting against the action of free radicals [11]. Many epidemiological studies have shown that the consumption of leafy plant vegetables containing phenolic and flavonoid compounds with potent antioxidant activity are associated with a lower incidence of cardiovascular diseases, cancer, diabetes and neurodegenerative diseases [12]. Wild edible leafy plants are primary sources of dietary requirements utilised by the native communities of Nepal as medicine, salad, juice or pickle [13,14]. Although some research on wild edible plants has been documented from Nepal, it is still limited to the survey of traditional utilisation among local people [15]. The potential phytochemical characteristics and antioxidant activity for these plants from Nepal have not been reported to date. Eight traditionally used wild edible leafy plants (Table 1) from Nepal were selected to evaluate the phenolic and flavonoid content along with the antioxidant activity for the first time.

## 2. Results and Discussion

Wild edible plants have remarkable roles in and contributions to Nepalese diets and food security. The utilisation and knowledge of wild vegetables as a nutritional source is confined to local people. A detailed literature review into the phenolic and flavonoid content of the wild vegetables consumed in the Nepalese diet including their antioxidant activity has not been carried out to date [16]. This study appears to be the first to validate the phenolic and flavonoid content, as well as the antioxidant efficacy of methanolic extracts of selected plants from Nepal.

### 2.1. Total Phenolic Content

Phenolic compounds are important plant constituents with redox properties responsible for antioxidant activity [22]. The hydroxyl groups in plant extracts are responsible for facilitating free radical scavenging. As a basis, phenolic content was measured using the Folin–Ciocalteu reagent in each extract. The results were derived from a calibration curve (y = 9.53x − 0.13, R^2^ = 0.996) of gallic acid (0–250 µg/mL) and expressed in gallic acid equivalents (GAE) per gram dry extract weight (Table 2). The content of phenolic compounds in methanol extracts ranged from 292.65 to 72.66 mg GAE/g, representing an approximate four-fold variation. *A. sessilis*, *C. tora* and *P. oleracea* had the greatest phenolic contents (292.65 ± 0.42, 287.73 ± 0.16 and 216.96 ± 0.87 mg GAE/g, respectively), while the smallest phenolic contents were found in *B. alba, I. aquatica*, *and S. nigrum* (72.66 ± 0.46, 77.06 ± 0.70 and 97.96 ± 0.62 mg GAE/g, respectively).

The extraction procedures and solvents are responsible for dissolving the endogenous compounds of the plants [23]. Moreover, plant components can be polar or non-polar in nature. Phenolic compounds are more soluble in polar organic solvents due to the presence of a hydroxyl group, therefore methanol was selected as the extracting solvent [24]. Comparing the works of literature, Lee et al. reported a total phenol content (TPC) of 56.8 ± 5.9 mg GAE/g fresh weight of *A. sessilis* and 36.4 ± 6.1 mg GAE/g fresh weight of *I. aquatica* [6] in acetone–water–acetic acid extracts. Adebooye et al. found a TPC 0.704 mg GAE/g fresh weight of *S. nigrum* in a water extract [12]. Yen et al. found a TPC of 180.64 ± 6.51 mg GAE/g in water extracts of *C. tora* [11]. Uddin et al. found a TPC of 3.6 ± 0.089 mg GAE/g dry weight in the methanol extract of *P. oleracea* [25]. The values of phenolic content in this current study varied slightly compared to those in the literature. This may be due to the presence of different amounts of sugars, carotenoids or ascorbic acid, or the duration, geographical variation or methods of extraction, which may alter the amount of phenolics [26].

### 2.2. Total Flavonoid Content

As a basis quantitative determination, flavonoid contents in selected plant extracts were determined using aluminium chloride in a colorimetric method. The results were derived from the calibration curve (y = 0.0057 + 0.0127, R^2^ = 0.9973) of quercetin (0–100 µg/mL) and expressed in quercetin equivalents (QE) per gram dry extract weight (Table 2). The flavonoid content in methanol extracts ranged from 37.86 to 6.61 mg QE/g, representing an approximate six-fold variation. *P. oleracea*, *C. tora, and L. cephalotes* had the greatest flavonoid content (39.38 ± 0.57, 37.86 ± 0.53 and 36.95 ± 0.44 mg QE/g respectively), while the smallest amounts of flavonoids were found in *I. aquatica, B. alba and S. nigrum* (6.61 ± 0.42, 6.97 ± 0.62 and 16.42 ± 0.39 mg QE/g respectively).

Flavonoids are secondary metabolites with antioxidant activity, the potency of which depends on the number and position of free OH groups [27]. In a survey of past literature reports it was found that Kumar et al. reported a TFC of 21.53 mg QE/g dry weight in the methanol extract of *C. tora* [28], Adebooye et al. determined a TFC of 0.64 mg catechin equivalents per gram fresh weight in the water extract of *S. nigrum*, and Uddin et al. found the TFC of 49.2 ± 3.4 mg rutin equivalents per gram dry weight in the methanol extract of *P. oleracea* [13]. As reported in the literature, genetic diversity and biological, environmental, seasonal and year-to-year variations significantly affected the flavonoid content of vegetables [28].

### 2.3. DPPH Radical Scavenging Activity

The DPPH radical scavenging activities of selected medicinal plants are presented in Figure 1. All the plant extracts showed concentration-dependent increases in radical scavenging capacity. The greatest DPPH radical scavenging potency of with a minimum IC_50_ value was recorded for *C. tora* (9.898 µg/mL), followed by *L. cephalotes* (33.82 µg/mL), *A. sessilis* (35.39 µg/mL), *P. oleracea* (41.18 µg/mL), *D. muricata* (41.58 µg/ mL), *S. nigrum* (42.38 µg/mL), *I. aquatica* (42.43 µg/mL) and *B. alba* (45.68 µg/mL). All data were compared with the IC_50_ value of standard ascorbic acid (3.276 µg/mL), as presented in Table 2.

DPPH is a stable organic free radical, which loses its absorption spectrum band at 515–528 nm when it accepts an electron or a free radical species [29]. The DPPH assay is a simple, acceptable and most widely used technique to evaluate the radical scavenging potency of plant extracts [30]. The antioxidants are the components of the plants which are capable of enacting the visually noticeable quenching of the stable purple-coloured DPPH radical to the yellow-coloured DPPH [31].

### 2.4. Hydrogen Peroxide Scavenging Activity

The H_2_O_2_ scavenging potency of the methanol extract of selected plants was evaluated and presented in Figure 2. All of the plants show a concentration-dependent increase in radical scavenging properties. The greatest radical scavenging potency of with minimum IC_50_ value was recorded for *L. cephalotes* (16.25 µg/mL), followed by *S. nigrum* (17.89 µg/mL), *I. aquatica* (19.86 µg/mL), *C. tora* (22.52 µg/mL), *A. sessilis* (22.74 µg/mL), *P. oleracea* (24.37 µg/mL), *B. alba* (28.88 µg/mL), and *D. muricata* (29.22 µg/mL). All data were compared with an IC_50_ value of standard ascorbic acid (16.26 µg/mL) as presented in Table 2.

Hydrogen peroxide (H_2_O_2_) is a strong oxidizing agent, which can activate the signalling pathway to stimulate cellular proliferation [32], or differentiation [33]. It is generated in a biological system by many oxidizing enzymes such as superoxide dismutase [34]. However, aberrant accumulation of H_2_O_2_ is responsible for oxidative stress and inflammation reactions, which are correlated with pathological conditions like cancer, diabetes, and cardiovascular diseases [35,36]. This is because of rapid decomposition of H_2_O_2_ and subsequent generation of the hydroxyl radical (•OH) that initiates lipid peroxidation and damage of cellular components [37]. Regulation of H_2_O_2_ generation by plant antioxidants is of high interest in biological research.

### 2.5. Ferric Reducing Antioxidant Power (FRAP) Assay

The reducing power of Fe^2+^ by selected plants was evaluated (Figure 3). Like the radical scavenging activity, all of the extracts from the selected plants showed concentration-dependent reducing power. The greatest reducing antioxidant power was recorded for *A. sessilis*, followed by *C. tora, P. oleracea, L. cephalotes, I. aquatica, B. alba, D. muricata and S. nigrum* compared to standard ascorbic acid.

The transformation ability of compounds from Fe^3+^/ferricyanide complex to Fe^2+^/ferrous form acts as a potential indicator for antioxidant activity [38]. In the FRAP assay, the yellow colour test solution changes to green and blue depending on the reduction capacity of extracts or compounds [39,40]. The presence of reductants in the test solution reduces Fe^3+^ to Fe^2+^, which can be monitored by measurement of Perl’s Prussian blue colour at 700 nm [41]. The FRAP assay of antioxidants is convenient, reproducible and linearly concentration-dependent [42].

### 2.6. Ferric Thiocyanate in a Linoleic Acid System

Lipid peroxidation in a biological system under oxidative stress produces lipid hydroperoxides, which further transform to lipid alkoxyl (LO•) or lipid peroxyl (LOO•) radicals. These radicals are involved in great amounts of cellular damage, inducing degenerative diseases [43]. Lipid hydro-peroxides are stable at room temperature, but they are decomposed to radicals by heat, UV light or by transition metals [44]. The antioxidant activity of the plant extracts was determined by peroxidation of linoleic acid using the thiocyanate method at 37 °C after the addition of 100 µg/mL of extract sample. During lipid peroxidation, peroxides are generated which oxidize Fe^2+^ to Fe^3+^ upon the addition of FeCl_2_. On the addition of thiocyanate (SCN^−^), it gives a ferric thiocyanate complex with maximum absorbance at 500 nm.

The antioxidant effect of selected plant extracts in preventing the peroxidation of linoleic acid as measured by the ferric thiocyanate method is represented in Figure 4. In the control, the absorbance increased to 2.14 ± 0.02 at 72 h, then decreased. This was due to the formation of secondary oxidation products, which stop peroxide formation [45]. In the presence of antioxidants, the oxidation of linoleic acid will be slow, and the colour development from thiocyanate will be low. Of the selected plant extracts, minimum absorbance was observed for *C. tora* with minimum peroxide formation at observed time intervals. Like other antioxidant activity, extracts with greater phenolic or flavonoid contents showed lower absorbance due to minimum peroxidation.

### 2.7. The Correlation between the Total Phenolic and Flavonoid Content, and the Antioxidant Activity

Phenolic and flavonoid molecules are important antioxidant components which are responsible for deactivating free radicals based on their ability to donate hydrogen atoms to free radicals. They also have ideal structural characteristics for free radical scavenging [41]. Different literature reports indicate a linear correlation of total phenolic and flavonoid content with antioxidant capacity [13].

The correlation of total phenolic and flavonoid content with antioxidant capacity is shown in Figure 5a,b. High correlations between antioxidant capacity and total phenols (DPPH, R^2^ = 0.75; H_2_O_2_, R^2^ = 0.71) and total flavonoids (DPPH, R^2^ = 0.84; H_2_O_2_, R^2^ = 0.66) were observed at a 95% confidence level. By comparing the correlation coefficients (R-values), it is possible to suggest that phenolic and flavonoid groups are highly responsible for the antioxidant activity of the selected plant extracts.

## 3. Materials and Methods

### 3.1. Chemicals and Drugs

Quercetin, gallic acid, and DPPH was procured from Sigma Aldrich, India. Folin–Ciocalteu reagents and ascorbic acid were purchased from S. D. Fine Chem Limited, India. Aluminium chloride, trichloroacetic acid, ferric chloride, and potassium ferricyanide were purchased from Ranchem, India. Linoleic acid was procured from Acme Synthetic Chemicals, India. All chemicals and solvents used were of analytical grade.

### 3.2. Preparation of Methanolic Extracts

Eight wild species of mature green leafy plants (Table 1) were collected from the Rupandehi, Nepal (27°30’41.6” N, 83°21’01.6” E) and further identified by Professor Subodh Khanal (Botanist, Department of Environmental Sciences, Institute of Agriculture and Animal Sciences, Tribhuvan University). The aerial parts of the collected plants were dehydrated and pulverized. The grounded powders (70 g) were immersed in methanol (350 mL) for 7 days at room temperature with frequent agitation. The extracts were filtered using a Buckner funnel and Whatman No. 1 filter paper. Each filtrate was concentrated to dryness in a rotary evaporator (Büchi Labortechnik, Germany) under reduced pressure and controlled temperature (40–50 °C) to give final extracts, which was stored at 4 °C in an airtight container until further use.

### 3.3. Determination of Phenolic Contents

The total phenolic content was determined for individual extracts using the Folin–Ciocalteu method [6]. Briefly, 1 mL of extract (100–500 µg/mL) solution was mixed with 2.5 mL of 10% (w/v) Folin–Ciocalteu reagent. After 5 min, 2.0 mL of Na_2_CO_3_ (75%) was subsequently added to the mixture and incubated at 50 °C for 10 min with intermittent agitation. Afterwards, the sample was cooled and the absorbance was measured utilizing a UV Spectrophotometer (Shimazu, UV-1800) at 765 nm against a blank without extract. The outcome data were expressed as mg/g of gallic acid equivalents in milligrams per gram (mg GAE/g) of dry extract.

### 3.4. Determination of Flavonoid Contents

The flavonoid contents of individual extracts were measured as per the Dowd method [46]. An aliquot of 1 mL of extract solution (25–200 µg/mL) or quercetin (25–200 µg/mL) were mixed with 0.2 mL of 10% (w/v) AlCl_3_ solution in methanol, 0.2 mL (1 M) potassium acetate and 5.6 mL distilled water. The mixture was incubated for 30 min at room temperature followed with the measurement of absorbance at 415 nm against the blank. The outcome data were expressed as mg/g of quercetin equivalents in milligrams per gram (mg QE/g) of dry extract.

### 3.5. DPPH Radical Scavenging Activity

The radical scavenging activity (RSA) of the crude extracts was adopted to measure antioxidant activity using the DPPH method [47]. Briefly, 2 mL of extract solution (1–100 µg/mL) in methanol was added to 2 mL of DPPH (0.1 mM) solution. The mixtures were kept aside in a dark area for 30 min and absorbance was measured at λ_max_ 517 nm against an equal amount of DPPH and methanol as a blank. The percentage of DPPH• scavenging (RSA %) was estimated using the equation:% scavenging of DPPH• = [(A_0_ − A_1_)/A_0_] × 100,(1)
where A_0_ = absorbance of the control and A_1_ = absorbance of the test extracts.

### 3.6. Hydrogen Peroxide Scavenging Activity

The radical scavenging activity of individual extracts was determined using the H_2_O_2_ method. [48]. Briefly, 2 mL of extract solution (10–100 µg/mL) in methanol was added to 4.0 mL of H_2_O_2_ (20 mM) solution in phosphate buffer (pH 7.4). After 10 min, the absorbance was measured at λ_max_ 230 nm against the phosphate buffer blank solution. The percentage scavenging of H_2_O_2_ was calculated using the equation:% scavenging of H_2_O_2_ = [(A_0_ − A_1_)/A_0_] × 100,(2)
where A_0_ = absorbance of the control (phosphate buffer with H_2_O_2_) and A_1_ = absorbance of the test extracts.

### 3.7. Ferric Reducing Antioxidant Power (FRAP) Assay

The reducing powers of the individual extracts that reflected their antioxidant activity were determined using the modified Fe^3+^ to Fe^2+^ reduction assay [49]. Briefly, 1 mL of extract solution (10–200 µg/mL) in methanol was added to 2.5 mL of 0.2 M sodium phosphate buffer (pH 6.6) and 2.5 mL of 1% (w/v) potassium ferricyanide (K_3_Fe(CN)_6_) solution. The mixture was vortexed and incubated at 50 °C for 20 min assisted with a vortex shaker followed by the addition of 2.5 mL 10% (w/v) trichloroacetic acid and centrifugation at 3000 rpm. Finally, 2.5 mL of the supernatant was mixed with 2.5 mL deionized water and 0.5 mL 0.1% (w/v) ferric chloride, and Perl’s Prussian blue colour was measured at λ_max_ 700 nm against a blank. Increased absorbance of the reaction mixture indicated greater reducing power.

### 3.8. Ferric Thiocyanate (FTC) in a Linoleic Acid System

The antioxidant activity of selected plant extracts was determined by a linoleic acid system as described by Zou et al. [40]. Firstly, a linoleic acid emulsion was prepared by mixing and homogenizing 0.5608 g linoleic acid, 0.5608 g of Tween 20 emulsifier and 100 mL phosphate buffer (0.2 M, pH 7.0). Each individual extract solution (1 mL, 100 µg/mL) in ethanol was mixed with 5.0 mL linoleic acid emulsion (0.2 M, pH 7.0) and 5 mL phosphate buffer (0.2 M, pH 7.0). The peroxidation reaction was accelerated by incubating the mixture in a dark room at 37 °C. The peroxide level was measured by reading the absorbance at 500 nm on a spectrophotometer as per the thiocyanate method by mixing 10 mL ethanol (75%), 0.2 mL ammonium thiocyanate (30% w/v), and 0.2 mL FeCl_2_ (2 mM in 3.5% HCl). On the other hand, the control sample was prepared by mixing 5 mL linoleic acid emulsion and 5 mL phosphate buffer. Increased absorbance of the reaction mixture indicated higher linoleic acid peroxidation.

### 3.9. Statistics

The data were reported as the mean ± standard deviation. Linear regression coefficient (R^2^) for phenolic and flavonoid content with antioxidant activity was analysed by Graph Pad Prism for Windows, Version 7 (Graph Pad Software, San Diego, CA, USA). A *p*-value < 0.05 was considered significant.

## 4. Conclusions

In this study, the assessment of antioxidant activity indicates that edible wild leafy plants with higher phenolic and flavonoid contents could be a significant source of natural antioxidants. Although the parameters used in this study were not disease-specific, the quantification of antioxidant properties can serve as a guide for the use of these plants for ROS-related diseases. The selected plants with high antioxidant activity might be proposed for impeding toxic oxidation in nutraceuticals or drugs for the treatment of coronary diseases. Further investigation into the isolation and identification of responsible antioxidant components and their mechanism of action is necessary to better understand their ability to control diseases that have a significant impact on quality of life.

## Figures and Tables

**Figure 1 plants-08-00096-f001:**
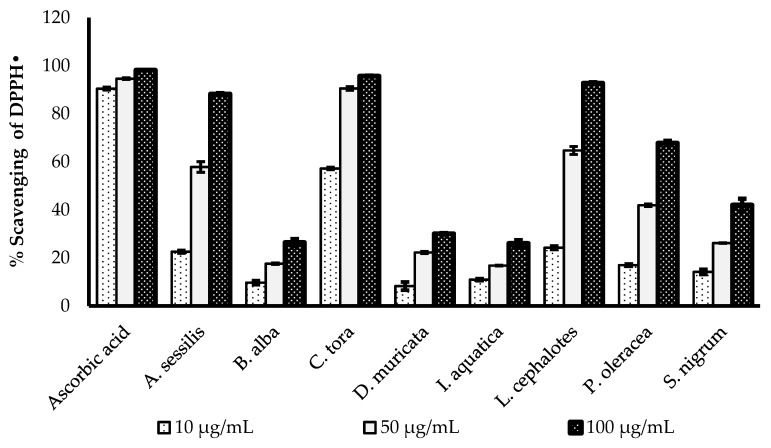
Comparison of DPPH• scavenging activity of ascorbic acid and selected wild leafy plants. Results expressed as the mean ± standard deviation (n = 3) at concentrations of 10, 50 and 100 µg/mL.

**Figure 2 plants-08-00096-f002:**
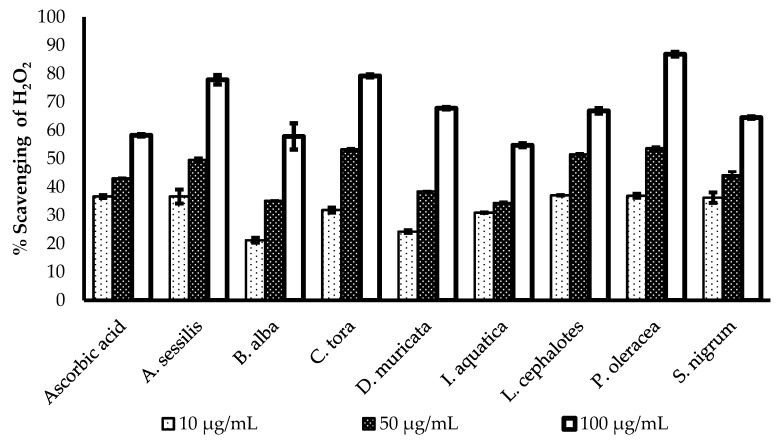
Comparison of H_2_O_2_ radical scavenging activity of ascorbic acid and selected plants extracts. Results expressed as the mean ± standard deviation (n = 3) at concentrations of 10, 50, and 100 µg/mL.

**Figure 3 plants-08-00096-f003:**
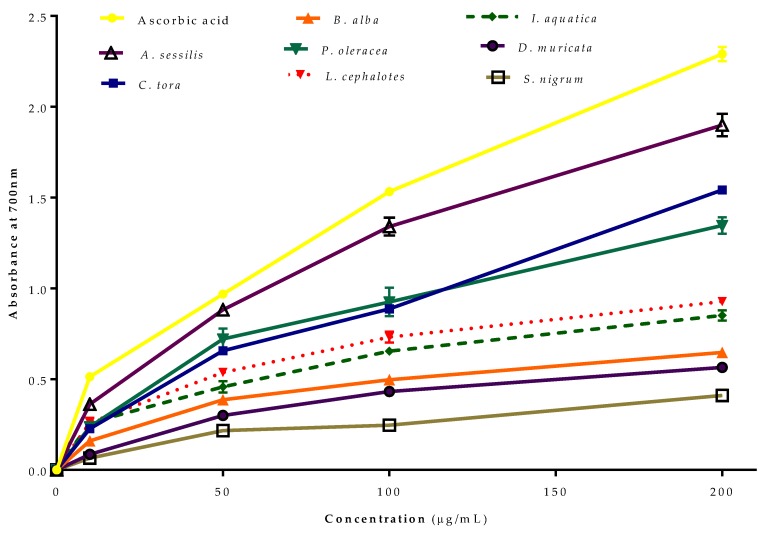
Ferric reducing antioxidant power (FRAP) of selected wild leafy plants. Results expressed as the mean ± standard deviation (n = 3) at concentrations of 10–200 µg/mL.

**Figure 4 plants-08-00096-f004:**
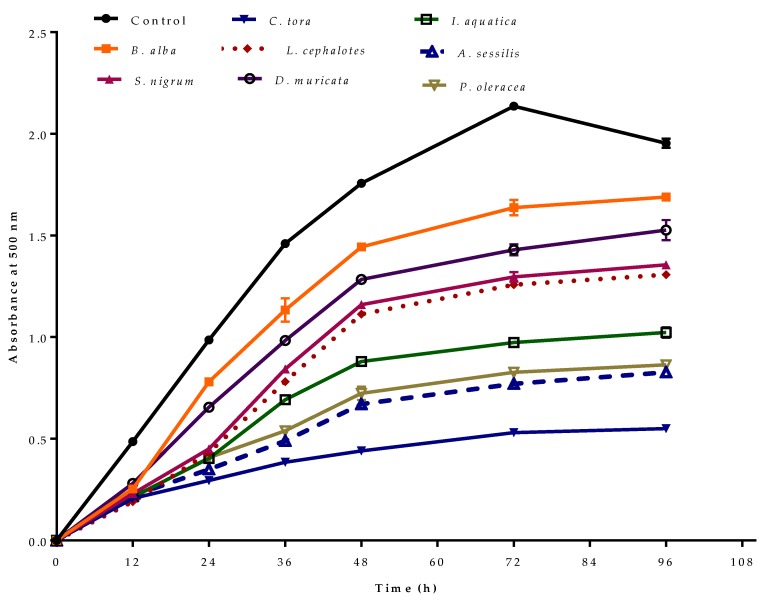
Antioxidant activity of selected wild leafy plants by ferric thiocyanate method–linoleic acid system at different time intervals. Results expressed as the mean ± standard deviation (n = 3) at a concentration of 100 µg/mL.

**Figure 5 plants-08-00096-f005:**
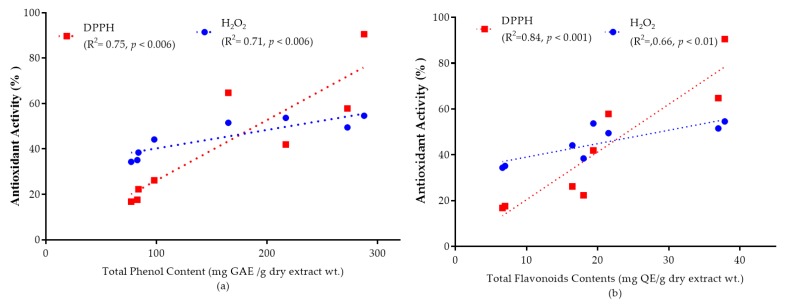
Graphs represent the mean value of antioxidant activity (%) at 50 mg/mL extract solution versus (**a**) total phenolic content and antioxidant activity; (**b**) total flavonoid content and antioxidant activity. The correlation coefficient values for total phenolic (DPPH, R^2^ = 0.75; H_2_O_2_, R^2^ = 0.71) and total flavonoid contents (DPPH, R^2^ = 0.84; H_2_O_2_, R^2^ = 0.66) was observed at a 95% confidence level.

**Table 1 plants-08-00096-t001:** Scientific names, voucher numbers, local names, parts used, and medicinal uses of selected wild leafy plants.

Scientific Names [Voucher No.]	Local Names	Parts Used	Medicinal Uses
*Alternanthera sessilis* [UHS1707]	Bhiringi jhar	Whole Plants	Wounds, venereal disease, menstrual disorder, fever and bloody dysentery [16]
*Basella alba* [UHS1701]	Poi sag	Apical shoots	Insomnia [17]
*Cassia tora* [UHS1705]	Sano tapre	Leaves and seeds	Skin disease, gastrointestinal disorders [18]
*Digera muricata* [UHS1708]	Lehasuwa	Leaves and shoots	Urinary tract infection [19]
*Ipomoea aquatic* [UHS1703]	Kalami sag	Leaf and young buds	Ring worm and skin diseases [16]
*Leucas cephalotes* [UHS1706]	Drona puspi	Plant juice	Urinary complaints [20]
*Portulaca oleracea* [UHS1702]	Kulfa sag	Leaves, fruits and seeds	Blood purification, dental problems [18], antidiabetic [19]
*Solanum nigrum* [UHS1704]	Kaalo Bihin	Roots and fruits	Easy child delivery, intermittent fever [17,21]

**Table 2 plants-08-00096-t002:** Total phenolic and flavonoid contents of selected wild leafy plants (*n* = 3).

Plant Sample	TPC (mg GAE/g dry extract wt)	TFC (mg QE/g dry extract wt)	DPPH• scavenging IC_50_ (µg/mL)	H_2_O_2_ Scavenging IC_50_ (µg/mL)
Ascorbic acid	-	-	3.276 *	16.26 *
*Alternanthera sessilis*	292.65 ± 0.42	21.51± 0.46	35.39	22.74
*Basella alba*	72.66 ± 0.46	6.97 ± 0.62	45.68	28.88
*Cassia tora*	287.73 ± 0.16	37.86 ± 0.53	9.898	22.52
*Digera muricata*	83.69 ± 0.46	18.00 ± 0.68	41.58	29.22
*Ipomoea aquatica*	77.06 ± 0.70	6.61 ± 0.42	42.43	19.86
*Leucas cephalotes*	164.96 ± 0.67	36.95 ± 0.44	33.82	16.25
*Portulaca oleracea*	216.96 ± 0.87	39.38 ± 0.57	41.18	24.37
*Solanum nigrum*	97.96 ± 0.62	16.42 ± 0.39	42.89	17.89

TPC: total phenol content; TFC: total flavonoid content; GAE: gallic acid equivalents; QE: quercetin equivalents; wt: weight; DPPH•: DPPH radical; * reference values for ascorbic acid.

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
