# Peer review of "Total Phenolic Content, Flavonoid Content and Antioxidant Potential of Wild Vegetables from Western Nepal"

_plants, 2019, doi:10.3390/plants8040096_

Round 1

Reviewer 1 Report

p { margin-bottom: 0.1in; line-height: 115%; background: transparent none repeat scroll 0% 0%; }a:link { color: rgb(0, 0, 128); text-decoration: underline; }

Dear Authors,

In my opinion, the manuscript in its current form is incomplete. Please address my suggestions, that can be found below.

1.,  I think the novelty of your paper - in its current form - does not justify for publication in Plants. The only conclusion that can be drawn from the results is that the examined plants have in vitroantioxidant capacity, that is most likely due to their flavonoids.

However, work on in vitro anti-oxidant capacity has been already carried out in for the examined plants in the literature. In several cases, in vivo antioxidant studies are also available. Some articles: 10.5897/JMPR2013.2567 (A. sessilis), 10.1016/j.jep.2010.04.036 (B. alba), 10.1016/j.lwt.2006.05.010 (C. tora), 10.1016/j.jep.2008.12.006 (D. muricata, other in vivostudies were reported), 10.1002/jsfa.2125 (I. Aquatica, chemical characterization), 10.1016/j.jep.2009.09.042 (L. cephalotes, in vivo study); 10.1155/2014/951019 (P. oleracea, review).

As there is a multi-correlation among different antioxidant assays, testing the plants with several ones does not add new insights.

Extension possibilities include: 1., In-depth chemical characterization of the plants via putative structure elucidation via LC-MS, or, purification of the compounds and analysis by MS, and NMR, and optionally, CD. 2.,In vivo studies of the best plant in animal studies.

Minor issues.

1., Do not connect dots in Figs. 1-2, the series of the plants do not represent a kinetic or similar.

2., R2 ~ 0.66 and 0.71 is not “strong” correlation in my opinion. What are the confidence intervals?

Author Response

Thank you for your valuable comments and suggestions. 

We would like to clarify that the work done by us has been the first research work performed taking the wild vegetable from western region of Nepal. Detailed literature research into the phenolic and flavonoid content with their antioxidant activity present in wild vegetables consumed in Nepalese diet was not carried out yet. This is the first study to validate the phenolic, flavonoid content and antioxidant efficacy of selected plants methanol extract from Nepal. This research work is essentially significant for Nepalese farmers, Pharmaceuticals and nutraceuticals which can be further expanded to reach the outside world. No doubt the antioxidant capacity of the plants is mostly due to the flavonoids present in the plants which is equally supported by our findings as well and literatures. 

Finally we need to be honest that we do not have the extended facilities for in vivo animal models testing to find the efficacy of the extracts in live animals. Definitely structural elucidation using NMR would be more interesting. We were able to perform the experiments for which we have the facilities this time and for remaining research like our honorable reviewer suggested, we are thinking to carry after we get proper facilities. 

Thank you honorable reviewer for your valuable suggestions.

Reviewer 2 Report

Dear Editor, in the manuscript plants-4536847 authors measured antioxidant potential by using DPPH scavenging, hydrogen peroxide (H2O2), ferric reducing antioxidant power (FRAP) and ferric thiocyanate (FTC) methods and total phenolic and flavonoid contents in eight wild edible leafy plants from Nepal. In general, the experiment was well performed and the manuscript well written and thus, it could be suitable for publication in this journal.

Nevertheless, the following comments should be considered:

- Line 52: It should be “antioxidants are” instead of “is.”

- Line 55: It should be “content” instead of “composition“, because individual phenolics were not quantified.

- Line 60: The same as above.

- Lines 79-84: These values cannot be compared if they are not expressed in the same units. It should be better to express all of them in mg/100 fresh weight. These data could be added to Table 1 or be commented in the text because these units are more familiar to readers for comparative purposes and gives us a clear idea about if these values are higher than total phenolic concentration found in other plant species.

- Line 93: It should be “flavonoids” instead of “phenolics.”

- Figures 3 and 4: Values for Absorbance higher than 1 are not appropriate.

- Line 205: Was the whole plant used?

- What was the physiological stage at which the plant species were harvested?

Author Response

- Line 52: It should be “antioxidants are” instead of “is.”

Answer to the comments: As suggested by the reviewer we have changed the sentence which is marked by red color font in the revised manuscript. Thank you for your valuable suggestion.

- Line 55: It should be “content” instead of “composition“, because individual phenolics were not quantified.

Answer to the comments: As suggested by the reviewer we have changed the sentence which is marked by red color font in the revised manuscript. Thank you for your valuable suggestion.

- Line 60: The same as above.

Answer to the comments: As suggested by the reviewer we have changed the sentence which is marked by red color font in the revised manuscript. Thank you for your valuable suggestion.

- Lines 79-84: These values cannot be compared if they are not expressed in the same units. It should be better to express all of them in mg/100 fresh weight. These data could be added to Table 1 or be commented in the text because these units are more familiar to readers for comparative purposes and gives us a clear idea about if these values are higher than total phenolic concentration found in other plant species.

Answer to the comments: As suggested by the reviewer we have used the same units, which is marked by red color font in the revised manuscript. Thank you for your valuable suggestion.

- Line 93: It should be “flavonoids” instead of “phenolics.”

Answer to the comments: As suggested by the reviewer we have change the sentence which is marked by red color font in the revised manuscript. Thank you for your valuable suggestion.

- Figures 3 and 4: Values for Absorbance higher than 1 are not appropriate.

Answer to the comments: We are thankful to the reviewer for the comment but we do have many instances and articles already published by MDPI, Elsevier and Springer that the absorbance values can be higher than 1. We have worked out in accordance to the international practices hence we think there should be no problem with the values that are higher than 1.

Please find the articles for the reference as listed below.

doi:

10.3390/antiox4020394 (MDPI-Antioxidents).

10.1007/s13197-011-0389-x (Springer-Journal of Food science and Technology).

10.1016/j.phrp.2013.09.003 (Elsevier- Osong Public Health Res Perspect).

- Line 205: Was the whole plant used?

Answer to the comments: As suggested by the reviewer we have mentioned the part of plants (i.e. aerial parts) used in the revised manuscript which is marked by red color font. Thank you for your valuable suggestion.

- What was the physiological stage at which the plant species were harvested?

Answer to the comments: As suggested by the reviewer we have mentioned the physiological stage (i.e. mature plants were taken for the experiment) during the harvesting of plants used and the sentence which is marked by red color font in the revised manuscript. 

Thank you for your valuable suggestions.

Reviewer 3 Report

In the abstract, include the number of wild vegetables studied.  Please write out the complete names of the plant species.

In order to educate readers unfamiliar with these wild vegetables, a table of the plants, their traditional preparation and use would be useful in the introduction.  Please include the entire scientific names, including the botanical authority in this table.

For readers unfamiliar with total phenolic contents, flavonoid contents or antioxidant capacities of these particular plants, it would be useful to include in the discussion total phenolic contents, flavonoid contents, antioxidant capacities of food plants more familiar to readers (e.g., spinach, lettuce, cilantro, parsley, collard greens, mustard greens).

Table 1 should be expanded to include the antioxidant activities (DPPH IC50, H2O2 IC50, and FRAP, including their respective standard deviations) as well as total phenolic and total flavonoids.  This will make it easier for readers to compare values.

The penultimate revised manuscript should be proof-read by an English-speaking technical editor.

Author Response

In the abstract, include the number of wild vegetables studied.  Please write out the complete names of the plant species.

Answer to the comments: As suggested by the reviewer we have mentioned full name of the plants species used in this study. The sentence is marked by red color font in the revised manuscript. Thank you for your valuable suggestion.

In order to educate readers unfamiliar with these wild vegetables, a table of the plants, their traditional preparation and use would be useful in the introduction.  Please include the entire scientific names, including the botanical authority in this table.

Answer to the comments: As suggested by the reviewer we have introduced table 1 in introduction part, Mentioning scientific name, parts use and their traditional medicinal value. The sentence is marked by red color font in the revised manuscript. Thank you for your valuable suggestion.

For readers unfamiliar with total phenolic contents, flavonoid contents or antioxidant capacities of these particular plants, it would be useful to include in the discussion total phenolic contents, flavonoid contents, antioxidant capacities of food plants more familiar to readers (e.g., spinach, lettuce, cilantro, parsley, collard greens, mustard greens).

Answer to the comments: Thank you for your valuable suggestions. We have tried our best to make the introduction and discussion parts as helpful, useful and understandable as possible. We believe our viewers and readers won’t have any problem to understand and implement our research article. We could compare the contents and capacities from the literatures already in the databases but comparing domesticated vegetables with wild may not be appropriate unless approved by international agencies like FDA as people may take this as a substitute for domesticated and farmed vegetables. Risk factors are associated.

Table 1 should be expanded to include the antioxidant activities (DPPH IC50, H2O2 IC50, and FRAP, including their respective standard deviations) as well as total phenolic and total flavonoids.  This will make it easier for readers to compare values.

Answer to the comments: As suggested by the reviewer we have mentioned antioxidant activities (DPPH IC50, and H2O2 IC50,) as well as total phenolic and total flavonoids in table 2 (previously table 1). The sentence is marked by red color font in the revised manuscript. Thank you for your valuable suggestion.

The penultimate revised manuscript should be proof-read by an English-speaking technical editor.

Answer to the comments: As suggested by the reviewer manuscript has been proof read for technical grammar corrections. The sentence is marked by red color font in the revised manuscript. Thank you for your valuable suggestion.

Thank you very much for your time and productive suggestions.

Round 2

Reviewer 1 Report

Dear Authors,

I regret to say but I still think your paper contains insufficient novelty for publication in a journal like Plants. My previous comments still apply.

Best regards.

Author Response

Answer to the comment: We are very much thankful to the reviewer for the comment. We have edited the manuscript introduction part to provide a clear proof for the novelty and significance of our work. The edited parts can be seen highlighted by red fonts color in the revised manuscript.

Although some research on wild edible plants has been documented from Nepal, it is still limited to the survey of traditional utilisation among inherent people [16]. The potential phytochemicals characteristics and antioxidant activity are yet unreported for these plants from Nepal. Based on traditional used eight wild edible leafy plants (Table-1) from Nepal were selected for the first time to evaluate the phenolic and flavonoids content along with the antioxidant activity.

We highly appreciate the reviewer’s effort in bringing this manuscript to the highest standard possible. Thank you very much.

Thank you to our honorable Reviewer.

Round 3

Reviewer 1 Report

Dear Authors,

I regret to say but I think my previous comments still apply.

Best regards